# SARS-CoV-2 from Urban to Rural Water Environment: Occurrence, Persistence, Fate, and Influence on Agriculture Irrigation. A Review

**Giuseppe Mancuso** [1,2,*] **, Giulio Demetrio Perulli** [1] **, Stevo Lavrnić** [1] **, Brunella Morandi** [1] **and Attilio Toscano** [1]

1 Department of Agricultural and Food Sciences, Alma Mater Studiorum-University of Bologna, Viale Giuseppe Fanin 44-50, 40127 Bologna, Italy; giulio.perulli@unibo.it (G.D.P.); stevo.lavrnic@unibo.it (S.L.); brunella.morandi@unibo.it (B.M.); attilio.toscano@unibo.it (A.T.)
2 CIRI FRAME-Interdepartmental Centre for Industrial Research in Renewable Resources, Environment, Sea and Energy, Alma Mater Studiorum-University of Bologna, Via Selmi 2, 40126 Bologna, Italy
* Correspondence: g.mancuso@unibo.it; Tel.: +39-051-209-6182

**Abstract:** The novel coronavirus disease (COVID-19), originating from China, has rapidly crossed borders, infecting people worldwide. While its transmission may occur predominantly via aerosolization of virus-laden droplets, the possibility of other routes of contagion via the environment necessitates considerable scientific consideration. SARS-CoV-2 viral RNA has been detected in the feces of infected persons, and studies also have reported its occurrence in wastewater and surface water bodies. Therefore, water may be a possible route of virus outbreaks. Agricultural irrigation is the largest use of water globally, accounting for 70% of water use worldwide. Ensuring adequate water quality within irrigation practices is fundamental to prevent harm to plants and soils, maintain food safety, and protect public health. This review aims to gather information on possible SARS-CoV-2 transmission routes within urban and rural water environments, looking into the detection, persistence, and fate of SARS-CoV-2. Based on published literature, the effect of current treatment technologies in wastewater treatment plants (WWTPs) on SARS-CoV-2 inactivation has also been investigated. Preliminary research efforts that concentrated on SARS-CoV-2 indicate that the risk of virus transmission from the aquatic environment may currently be non-existent, although a few studies have reported the presence of SARS-CoV RNA in soils, whereas there are still no studies on the detection of SARS-CoV-2 in crops.

**Keywords:** coronavirus; COVID-19; SARS-CoV-2; water reuse; reclaimed water; wastewater; agriculture irrigation

## 1. Highlights

(a) Evidence of viable SARS-CoV-2 RNA in wastewater has been used for COVID-19 surveillance.
(b) Standardized methodological protocols are needed for the accurate estimation of SARS-CoV-2 in wastewater.
(c) Data on the infectivity and survival of SARS-CoV-2 in wastewater and freshwater are limited.
(d) The role of WWTP units for SARS-CoV-2 deactivation is still unexplored.
(e) SARS-CoV-2 RNA in water environments might represent a risk of irrigation water contamination.
(f) It is necessary to investigate the eventual persistence of SARS-CoV-2 in crops.

## 2. Introduction

In December 2019, COVID-19 began with a viral outbreak in the city of Wuhan of central Hubei province of China [1]. The World Health Organization (WHO), along with

Chinese authorities, then commenced working together since a cluster of about 40 cases of pneumonia of unknown etiology was detected, with some of the patients being vendors or dealers in the Huanan seafood market. In the meantime, on 11 January 2020, China announced its first COVID-19-related death of a 61-year-old man, after being exposed to the seafood market of Wuhan [2]. COVID-19 infection has rapidly spread to the rest of China as well as many other countries, and the number of infected cases has continued to increase significantly every day [3], leading the WHO to declare the outbreak as a Public Health Emergency of International Concern on 30 January 2020 [2,4]. On 11 February 2020, the WHO introduced a name for the novel coronavirus disease: "COVID-19" [4]. On March 11th, the WHO announced COVID-19 as a global pandemic as, by then, about 114 countries were affected [4]. In 2020, the pandemic led to 81,477,457 confirmed cases, and 1,798,120 deaths worldwide [5]. Although the first SARS-CoV-2 vaccinations started in December 2020, the situation still raises concern due to the growing number of positive cases [5] and the identification of new SARS-CoV-2 variants [6].

SARS-CoV-2 has been identified as an enveloped, non-segmented, positive-stranded ribonucleic acid (RNA) virus belonging to the family Coronaviridae, within the order Nidovirales [7]. SARS-CoV-2 is generally responsible for respiratory and gastrointestinal infections, which might range from mild, self-limiting conditions to more severe disorders (i.e., viral pneumonia with systemic impairment) [8]. In the past, coronaviruses have already caused two large epidemics: severe acute respiratory syndrome (SARS) [9] and Middle East respiratory syndrome (MERS) [10]. Then, towards the end of 2019, a novel mutation of the coronavirus (categorized as SARS-CoV-2) was identified as the pathogen responsible for the COVID-19 respiratory illness.

COVID-19 is principally spread via respiratory tract with high infectivity. Droplet transmission is the predominant route of contagion from person to person [11]. However, there are other routes besides respiratory transmission. SARS-CoV-2 can remain infectious over time on surfaces of domestic and public items [12]. Individuals can be potentially infected if they touch their mouth, nose, and eyes with their hands after they have been in contact with such contaminated surfaces [13]. Furthermore, several studies reported SARS-CoV-2 shedding in human stool [14–16], and therefore in wastewater (WW). The detection of viable SARS-CoV-2 in WW raises concern of the possible spread of the COVID-19 disease through different environmental compartments [17].

The urgent need for research on SARS-CoV-2 in WW has led to the application of the wastewater-based epidemiology (WBE) approach, which aims to estimate the prevalence of COVID-19 in a given WWTP catchment population [18]. Its efficiency lies in the fact that WW contains human excreta with SARS-CoV-2 RNA within a given catchment and, therefore, it can be used as an early warning of disease outbreaks, as well as to evaluate the efficacy of public health interventions [19]. Ahmed et al. proposed [19] a mathematical formula to estimate the prevalence of SARS-CoV-2 infection by using a mass balance on the total concentration of SARS-CoV-2 RNA copies in WW each day, and the number of SARS-CoV-2 RNA copies shed in stool by an infected individual each day (Equation (1)):

$$IP = \frac{\frac{C}{V} \times \frac{V}{day}}{\frac{F}{person \times day} \times \frac{C}{F}} \tag{1}$$

where IP is the number of infected persons; C is the SARS-CoV-2 concentration (copies); V is the volume (L) of WW; *F* is the mass of feces (g). Since efficient use of WBE needs access to WW that is centrally collected, composited, and treated, the widespread use of WBE is actually limited to about 27% of the global population [20]. On the contrary, the remaining part of the population is not served by WW treatment facilities [20], confirming the possibility of SARS-CoV-2 environmental contamination and spread through WW. This situation is further exacerbated by the lack of in-depth studies in the literature on the infectivity and transmission of SARS-CoV-2 present in WW.

This review aims to provide a global understanding on how SARS-CoV-2 could enter the urban water cycle [17,21], and eventually move from urban to rural water environments, perhaps reaching agricultural crops. Ensuring the provision of water with a certain quality for irrigation purposes is fundamental for public health, especially when sustainable, safe, and cost-efficient strategies such as water reuse are promoted [22]. The United Nations, through the definition of the Sustainable Development Goal on water (SDG 6) [23], have recently emphasized the role of water reuse as an alternative source of water supply in order to cope with several issues associated with water scarcity and drought events, which are likely to be more severe and more frequent in the future due to climate change and increasing population [24]. With this in mind, an overview of current knowledge on the role of WWTPs for SARS-CoV-2 mitigation in WW and the eventual impact of the virus on agricultural irrigation will be described.

### 3. SARS-CoV-2 RNA Detection/Persistence in WW and Fate in WWTPs

SARS-CoV-2 RNA has been detected in stool samples of both symptomatic and asymptomatic infected persons [14,15,25–27], and when the infection was no longer detectable in their oral swab samples [28]. Viral SARS-CoV-2 RNA concentrations in feces have been generally observed to vary from $10^2$ to $10^8$ copies per gram of feces [29,30]. Many efforts have therefore been made by researchers to define suitable methods for the molecular detection of SARS-CoV-2 RNA in WW. The first attempts to measure SARS-CoV-2 in WW have been made worldwide, namely in the Netherlands [31], Australia [19], Spain [32], Germany [33], the USA [34], India [35], Japan [36], etc. However, the urgent need has immediately highlighted a lack of validated methodological protocols to follow for WW sample collection, storage, and concentration as well as for SARS-CoV-2 RNA extraction, detection, and quantification [37]. The number of SARS-CoV-2 RNA copies has been observed to remain surprisingly stable under different storage conditions (i.e., +4.0 °C, −20 °C, −75 °C for 29, 64, and 84 days, respectively) [38]. Several sample concentration methods have been adopted for the detection and quantification of SARS-CoV-2 RNA in WW, including electronegative membrane adsorption [39], ultra-filtration [19], ultra-centrifugation [39], polyethylene glycol (PEG) precipitation [39], and a combination thereof.

Experimental tests proved that for higher WW sample volumes (in the order of 500 mL), the detection of SARS-CoV-2 was not possible, probably due to the low viral load. On the contrary, preconcentrated WW samples (from 150 up to 200 μL) have showed minimal concentrations of $10^3$ copies $L^{-1}$, which in most of the cases has been identified as a threshold for SARS-CoV-2 RNA detection [40].

Once WW samples are concentrated, different processing methods and targeted genes have been proposed for SARS-CoV-2 RNA extraction, mainly exploiting reverse transcription quantitative polymerase chain reaction (RT-qPCR) analysis. In fact, RT-qPCR has allowed, also through the use of proper kits (i.e., TaqMan), the amplification of the genes encoding the proteins of the nucleocapsid (N1, N2, N3), which have been observed to be specific for SARS-CoV-2 RNA detection [41]. However, the main steps involved in the SARS-CoV-2 RNA extraction process have been cell lysis, denaturation of DNA and proteins, denaturation and inactivation of RNases, separation or removal of cellular components, and SARS-CoV-2 RNA recovery [37]. To date, no studies on the comparison of different SARS-CoV-2 extraction methods have been reported in the literature, although it has been observed that discordant measurements could be detected, since WW is a complex matrix with a high content of different organic and inorganic compounds, which could therefore be inhibitory to RT-qPCR analysis.

The definition of a univocal and optimized methodological framework concerning the detection and quantification of SARS-CoV-2 in WW is still underway in many geographical regions. In the meantime, the outcomes of the available studies have been expressed as (i) absence or presence of SARS-CoV-2 RNA and (ii) "number of gene copies" ÷ "sample volume". In the first case, results have been reported through RT-qPCR analysis, while, in the second case, a quantitative calibration curve has been exploited to compare measured

SARS-CoV-2 RNA concentrations in WW samples with already known concentrations of the virus. However, during the pandemic, most of the studies have referred only to the isolation and detection of SARS-CoV-2 RNA in raw WW. On the contrary, only a few studies involved infectivity tests of SARS-CoV-2 genes to investigate whether genetic material was present as intact virus particles or as free RNA fragments.

The first findings on coronavirus persistence in WW have shown its rapid death, with the time required for the virus titer to decrease by 99.9% ranging from 2 to 4 days [42]. However, the persistence of SARS-CoV-2 viral load in WW might depend on various factors. Temperature is one of them: SARS-CoV-2 viral titer was observed to decline more rapidly at 23 °C, and the reduction was much faster at 25 °C than at 4 °C [43]. On the contrary, some studies have reported the evident presence of SARS-CoV-2 viral genome in untreated WW even at high ambient temperatures [35]. Among the other factors, the level of organic matter and presence of antagonistic bacteria can influence the survival of SARS-CoV-2 in WW [34]. In particular, suspended particles, including colloidal material, microbial communities, algae, and chemical or biological aggregates present in WW not only can serve as reservoirs for SARS-CoV-2, but also maintain its activity by defending it from the oxidizing agents prevailing in WW [42].

Apart from domestic households, WW treatment infrastructures can also receive SARS-CoV-2 from sources with increased concentrations, such as hospitals, community clinics, and nursing homes. Under these circumstances, Naddeo and Liu [17] recommended a fit for purpose and decentralized virus inactivation treatment for WW discharged from these places in order to reduce their virus load and a risk of secondary transmission. If, on one hand, the current need is to understand the actual capacity of SARS-CoV-2 to retain its infectivity in WW, on the other hand, it would also be useful to investigate its fate in WWTPs in order to eventually achieve its complete inactivation.

Concerning SARS-CoV-2 infectivity in WW, every possible attempt was made to guarantee the safety of operators in WWTPs, adopting all the necessary precautions to follow by personnel to prevent their exposure to WW. These include using standard protocols, safe work practices, and personal protective equipment regularly required for work activities when handling untreated WW. To date, no other additional COVID-19-specific protections are recommended for workers involved in WW management operations, including those at WWTP facilities [44].

Another fundamental challenge is to ensure the complete removal of SARS-CoV-2 in WW by the implementation of suitable treatment methods in WWTPs, since different studies have recently shown the presence of various viruses in both secondary-treated and disinfected WWTP effluents [45], thus highlighting the potential risk of disease outbreaks. Unfortunately, the fate of SARS-CoV-2 in WWTPs and its eventual removal during the different treatment stages is still unexplored and requires urgent attention, especially where treated effluent is utilized as reclaimed water.

The conventional WW treatment systems consist of different physical, chemical, and biological processes that are aimed at the removal of biodegradable organics, suspended solids, and pathogens from WW [46]. During the pandemic, attention was focused on SARS-CoV-2 detection in WW, thus mainly referring to WWTP influents. On the contrary, only a few studies have analyzed the presence of SARS-CoV-2 in WWTP effluents and at the different stages within WWTPs (Table 1).

**Table 1.** Prevalence of SARS-CoV-2 RNA in WWTPs.

| Country | Period | Wastewater | Detection Method | Treatment Process | $N_1/N_2$ (%) | C (Copies $L^{-1}$) | Removal (%) | Reference |
|---------|--------|-----------|------------------|-------------------|---------------|----------------------|-------------|-----------|
| Spain | March–April 2020 | WWTP influent | RT-qPCR | - | 35/42 (83%) | $2.5 \times 10^5$ | - | [31] |
| | | Secondary effluent | RT-qPCR | Activated sludge | 2/18 (11%) | $2.5 \times 10^5$ | 0% | |
| | | Tertiary effluent | RT-qPCR | Coagulation ↓ Flocculation ↓ Sand filtration ↓ Disinfection | 0/12 (0%) | 0 | 100% | |
| USA | January–April 2020 | WWTP influent | RT-qPCR | - | 2/7 (28%) | $4.9 \times 10^3$ | - | [33] |
| | | Secondary effluent | RT-qPCR | Activated sludge | 0/4 (0%) | 0 | 100% | |
| | | Tertiary effluent | RT-qPCR | Disinfection | 0/4 (0%) | 0 | 100% | |
| Japan | March–May 2020 | WWTP influent | RT-qPCR | - | 0/5 (0%) | 0 | - | [35] |
| | | Secondary effluent | RT-qPCR | Activated sludge | 1/5 (20%) | $2.4 \times 10^3$ | N/A | |
| India | May–June 2020 | WWTP influent | RT-PCR | - | 6/16 (37%) | - | - | [34] |
| | | Secondary effluent | RT-PCR | Moving bed biofilm reactor (MBBR) or sequencing batch reactor (SBR) | 0/1 (0%) | - | 100% | |
| | | Tertiary effluent | RT-PCR | Disinfection | 0/6 (0%) | - | 100% | |
| Germany | April 2020 | WWTP influent | RT-qPCR | - | 9/9 (100%) | $11 \times 10^1$ | - | [32] |
| | | Tertiary effluent | RT-qPCR | Ozonation | 4/9 (44%) | $19 \times 10^1$ | N/A | |

$N_1$ = Number of positive samples; $N_2$ = Number of collected samples; C = SARS-CoV-2 concentration (average values).

As shown in Table 1, in the study of Randazzo et al. [32], SARS-CoV-2 RNA was detected in 35 out of 42 influent samples with a concentration of $2.5 \times 10^5$ copies $L^{-1}$. Albeit in a smaller number of samples (two out of 18), the same concentration was detected after the activated sludge process, indicating that the secondary treatment was not completely able to remove SARS-CoV-2 from WW. In contrast, SARS-CoV-2 was not detected downstream of the tertiary treatments. This is in accordance with the study of Heramoto et al. [36], who also observed the presence of SARS-CoV-2 in secondary effluents. On the other hand, the studies of Sherchan [34] and Arora [35] did not find the presence of SARS-CoV-2 in either secondary or tertiary effluents. However, in the study of Westhaus

et al. [33], SARS-CoV-2 was also detected in tertiary effluents. Hence the need to better understand the fate of SARS-CoV-2 in the different units within WWTPs.

*Primary treatments* exploit physical processes such as screening, grit chamber, and primary sedimentation to remove suspended solids from WW. At this stage, SARS-CoV-2 viral particles can be removed after they have been adsorbed onto suspended solids, although removal rates should not exceed 25%, as confirmed by several pre-printed studies reported in the literature. Moreover, the viral load might persist within the settled sludge, entailing thus the risk of secondary contamination.

*Secondary treatments* consist of physico-biological processes (e.g., activated sludge, membrane bioreactors, sequencing batch reactors, secondary sedimentation, etc.) that allow the removal of biodegradable organic matter and suspended solids. The adsorption of the virus on suspended solids followed by settling in secondary clarifiers are the main removal mechanisms during the activated sludge process [42], although some of the studies reported to date in the literature show that the virus may still be present in secondary effluents (Table 1).

In previous investigations on other coronaviruses, their inactivation was found to be influenced by the level of organic matter and suspended solids, thereby allowing viruses to survive longer in primary-treated WW than in secondary-treated WW. Since in primary and secondary treatments, solid residues are largely removed, Westhaus et al. [33] have recently tried to distinguish SARS-CoV-2 concentrations for the aqueous and solid phases in both influent and effluent samples collected from WWTPs. When comparing the aqueous and solid phases, the authors observed higher SARS-CoV-2 RNA concentrations in the solid phase than in the aqueous phase, with the highest difference in the influents rather than effluents [33]. When comparing the aqueous phases only, the authors found higher SARS-CoV-2 concentrations in effluents than influents, probably due to the repartitioning of genetic material from the solid to the liquid phase during the WW treatment [33]. However, no differences between total SARS-CoV-2 concentrations (solid + liquid phases) in raw and treated WW were detectable, most likely due to the incompatibility among sample conditions and residence time of sludge in the WWTP units [33].

As well as for other viruses, the removal of SARS-CoV-2 through biological processes within the secondary stage of WWTPs can also be governed by several operating conditions (e.g., hydraulic retention time, biological solid retention time) and environmental parameters (e.g., temperature, pH) [42].

*Tertiary treatments* comprise physico-chemical processes to further reduce residual organic matter, turbidity, nutrients, and pathogens. Although several studies have reported virus genetic material both in primary- and secondary-treated effluents, when tertiary treatments were present in WWTPs, no genetic material was detected in the effluent stream (Table 1), except for the study of Westhaus et al. [33], in which, instead, the authors measured the presence of SARS-CoV-2 RNA after ozonation treatment (Table 1).

The enveloped structure of SARS-CoV-2 makes it vulnerable to external environmental conditions such as heat, pH, and reactive radicals, allowing different chemical disinfectants and physical agents to damage it. Chlorination has been by far the most used tertiary treatment method for WW disinfection [47]. It was suggested that SARS-CoV-2 in WW can be completely inactivated with chlorine (10 mg L$^{-1}$ for 10 min; free residue chlorine 0.4 mg L$^{-1}$) or chlorine dioxide (40 mg L$^{-1}$ for 30 min; free residue chlorine 2.19 mg L$^{-1}$) [43]. However, it has been proved that the use of chlorine as a disinfectant can lead to the formation of toxic and carcinogenic disinfection by-products originating from the reaction of chlorine with organic compounds present in WW [48]. Hence, chlorine has been replaced by alternative disinfectants, namely peracetic acid, sodium hypochlorite, and ozone [49].

As an alternative to the use of chemical reagents, ultraviolet light (UV) systems have been used for WW disinfection. In several pre-printed studies, electromagnetic energy (light) has been transferred to SARS-CoV-2 in WW, which was then inactivated. UV systems

are characterized by high removal efficiencies and do not generate by-products, but have high management costs [50].

Recently, advanced oxidation processes (AOPs) have been successfully implemented as tertiary treatment due to their high oxidation capability for a wide variety of organic/inorganic compounds, viruses, and bacteria, exploiting the chemical reactions promoted by hydroxyl radicals ($^\bullet$OH) [51]. Although this treatment technique seems to be very promising, it has not yet been tested for SARS-CoV-2 inactivation.

Membrane bioreactors (MBRs), which combine membrane-based filtration processes with suspended growth biological reactors, seem to be a valid alternative to guarantee the removal of SARS-CoV-2 from WW [35].

Phycoremediation is instead a novel, low-cost, and environmentally friendly technique that has been increasingly used, due to the involvement of microalgae or macroalgae for the removal of SARS-CoV-2 from WW [52].

Although the first outcomes on SARS-CoV-2 removal from WW seem to be reassuring, understanding the fate of SARS-CoV-2 in WWTPs is of major importance, with more research required to comprehend SARS-CoV-2 removal in the different treatment stages and, more significantly, where treated WW is used for irrigation purposes.

## 4. Potential Presence of SARS-CoV-2 in Irrigation Water

In the agricultural sector, irrigation is a fundamental practice to support plant growth and production when rainfall is not sufficient. As shown in Figure 1, freshwater might be contaminated by SARS-CoV-2, which could consequently enter the water cycle due to overflows in combined sewage systems [53], run-off events [54], and discharge of untreated WW into natural water bodies [55], representing a potential source of risk for food production, and thus for human health. It was suggested that SARS-CoV-2 can remain active for up to 25 days in water sources [56]. Therefore, it is important to evaluate possibilities and weak points in the water use chain in order to minimize the risks of transmission.

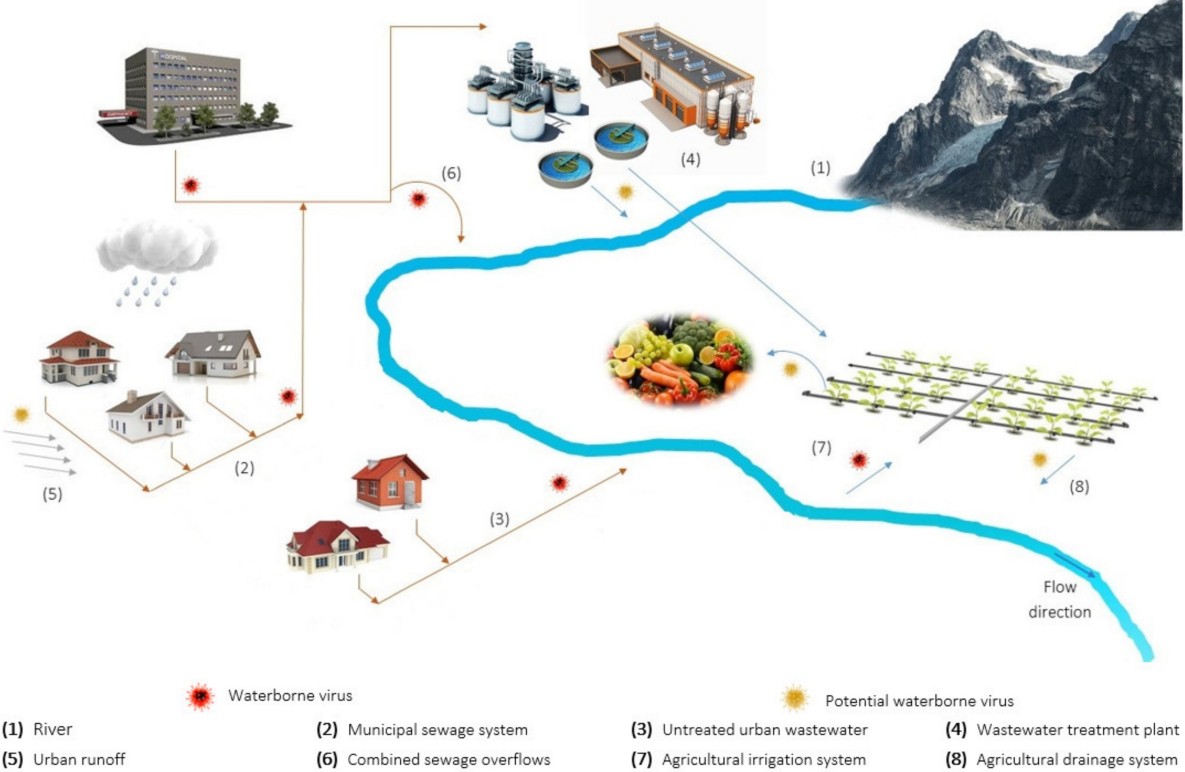

**Figure 1.** Schematic representation of possible SARS-CoV-2 pathways within the urban and agricultural water cycle.

Various coronaviruses have been found in both treated and untreated water, but it seems that the survival and sustainability of SARS-CoV-2 in aqueous environments depends on many factors, such as initial viral load, type of medium, exposure to sunlight, temperature, and organic matter. Coronaviruses can remain active and infectious in sewage and water for several days and weeks.

The direct reuse of wastewater is a valuable water source, and its use leads to a reduction of pressure on renewable water resources that are already scarce in some areas of the world [57]. However, before being reused, it is very important that in WWTPs suitable treatment methods are used, capable of avoiding the presence of SARS-CoV-2 within the effluents [58–60] (Figure 1). As already said in the previous sections, although secondary treatment can largely reduce, or even completely eliminate, SARS-CoV-2 from wastewater, its presence in WWTP effluent was confirmed. This can present particular danger for countries where wastewater treatment facilities are not present on a large scale or where they are outdated and under pressure due to the population increase and concentration in the urban centers. Moreover, particular attention should be paid to the cases when wastewater does not undergo any type of treatment before its reuse. The data indicate that less than 10% out of the total volume of wastewater used in irrigation worldwide receives some kind of treatment before its use [61]. Therefore, it was suggested that the authorities may promote micro-irrigation technologies that can prevent farmers' or fresh products' contact with irrigation water [62]. This might be even more important in this period when we still lack many data and certain results regarding SARS-CoV-2 transmission and risks. Even though there might be a theoretical risk of infection for farm workers that manage irrigation equipment and are in close contact with irrigation water, there is no evidence of such infection through aerosols [63].

SARS-CoV-2's presence in river waters was confirmed in some regions with a high prevalence of the disease. Moreover, the presence of coronaviruses from anthropogenic sources was confirmed in different surface water bodies [43], which was expected given the range of human activities that can affect water ecosystems. However, the infectivity of virus found in rivers in Italy was null and therefore preliminary results indicate that infection by river water seems improbable [64]. As for groundwater bodies, that are also often used for irrigation, currently, there is no evidence that human coronaviruses are present in them [43], and therefore they might be safer to use.

Another aspect to consider is that not all the supplied water for irrigation purposes is absorbed by agricultural crops, but a certain amount of it can enter the water cycle again and be discharged to water bodies. Agricultural drainage water cannot only be regarded as a source of diffuse pollution in aquatic ecosystems [65], but it might also be another way of possibly conveying SARS-CoV-2 to other environmental matrices (Figure 1). However, to the authors' best knowledge, currently, there is no evidence of such transport pathways in the scientific literature.

## 5. Potential Fate of SARS-CoV-2 in Agricultural Crops

Viral foodborne outbreaks are known to happen at different stages of the farm to fork chain of fresh products. One of the main transmission routes of food contamination could occur during the preharvest stage [66]. Water quality for irrigation represents a crucial issue for the safety of horticultural crops during preharvest management, especially when alternative water sources to freshwater (e.g., treated WW) are used for irrigational purposes.

The use of treated WW in agriculture is widely increasing, but less than 10% of collected WW worldwide receives any sort of treatment prior to its use in irrigation [61]. Hence, it is also very important to understand the fate of human pathogens, such as viruses at the plant level [62], especially where WW treatments (e.g., disinfection), before the irrigation stage, are inadequate or missing [67].

The literature reports how viruses can epiphytically survive from a few days to four weeks on plants that have been irrigated with treated or untreated WW, when direct contact has occurred between the water droplets and the plant tissue (though overcanopy/spray

irrigation methods). Viruses' survival mainly depends on their concentration in the reused water and on other environmental factors such as temperature variation due to sunlight irradiation [68,69]. Croci et al. [70] have found a significant number of viruses on the surface of plants irrigated with WW, which were able to survive under different environmental conditions through a variety of mechanisms, namely, adhesion to surfaces and internalization in fresh products, thus conditioning the effect of conventional processing and chemical sanitizing procedures that are usually adopted by the food industry.

The application of drip and subsurface irrigation methods represents a valuable approach to limit the viral contamination of edible products, since in these practices water usually does not come into direct contact with the plant canopy. However, even if the chances of food contamination are reduced, they are not completely undone due to the possible absorption of viruses through the root system. Based on that, it is clear that irrigation methods are already, per se, able to influence plant viral contamination with minimized risks where water is applied through drippers (e.g., directly to the soil without any contact with the canopy), while the risk is increased when it is delivered overcanopy (e.g., with fruit and leaf wetting) [66,71–73]. The adoption of micro-irrigation systems would then represent a safer strategy for limiting plants' viral contamination. However, researchers are trying to better understand the soil–root internalization processes [74].

Different studies, mostly related to horticultural crops (e.g., lettuce, tomato), demonstrated that human viral pathogen internalization is related to multiple factors, such as: plant species, phenological stage, abiotic and biotic stresses (e.g., cuticle and root wounds, superficial cuts, mechanical abrasions), growth substrate (soil and/or hydroponic), plant transpiration rate [75,76], and inoculum level [74,77–79]. However, plant viral internalization is a complex mechanism, and it is very difficult to identify those parameters that could facilitate plant viral uptake.

Among the other factors that have been mentioned above, the growth substrate plays a crucial role in the internalization of viruses, with a higher viral uptake in plants grown in hydroponic systems than in soil substrate [73,80]. The reduction of viral internalization when plants are grown in soil substrate seems directly related to the soil colloid particles (e.g., clays, organic matter), which bind to virus capsid ionic charge, lowering the risk of contamination [74,78].

However, viruses can be characterized by viral persistence, which mainly depends on soil conditions (temperature, moisture, pH), sunlight/UV radiation, and phosphorus and aluminum levels [81]. For instance, poliovirus, which is a non-enveloped virus, was found to differently persist in soil during summer (eleven days) than winter (ninety-six days) [69]. In the particular case of SARS-CoV-2, virus survival in soils could be affected by the presence of its viral envelope [7], which makes it more "susceptible" to death compared to non-enveloped viruses [82–84]. Furthermore, the possibility of SARS-CoV-2 to further mutate, due to its high mutation ability [6], should not be totally neglected when considering its potential persistence in soil conditions as well [85]. To date, research on SARS-CoV-2 survival on soils is still limited, with only a few studies dealing with eukaryotic viruses (humans viruses) or bacteriophages (bacterial viruses) [84].

Therefore, the COVID-19 pandemic has increased the concern about SAR-CoV-2 survival in soil environments. Recent research by Núñz-Delgado [86] indicates that soil could be polluted with SARS-CoV-2 if WW (especially if not adequately treated) is used, although no peer-reviewed studies have been reported in the literature on this possible viral persistence in soil.

When a viral load is present in the soil, its possible plant internalization is usually affected by the viral inoculum level and type of virus–plant interaction [74]. To date, limited research has been conducted on the mechanisms involved in root viral internalization [73]. Most of the studies were performed on human enteric viruses such as norovirus (NoV), which is considered among the major causes of food-borne outbreaks [87]. Unlike SARS-CoV-2, NoV is a non-enveloped single-stranded positive-sense RNA virus [88], resistant to

common disinfectants, with a low infectious dose and is highly stable and persistent in the environment (e.g., fresh and wastewater) [42,89–91].

Food contamination via roots mainly depends on the virus's ability to become internalized in the root system, and secondly on its ability to eventually translocate, through the vascular system, to the edible organ. The presence of internalized pathogens in the root system does not directly correlate with the presence of pathogens in the aboveground organs (e.g., fruit) [73]. DiCaprio et al. [74] stated that it is possible that viral translocation to seeds or fruits is less efficient than to other plant tissues (e.g., stems and leaves) where water flow in the xylem is faster due to the transpiration stream. As a consequence, Alum et al. [92] reported that the levels of viral contamination in root, stem, and leaves were higher than in the fruits. Enteric viruses are typically xylem driven from the root to the edible portions [79] and, thus, the transpiration rate of each tissue (e.g., leaf or fruit) could represent the cause of different levels of viral contamination. Therefore, crops where stems or leaves represent the edible sinks, such as vegetables like lettuce or fennel, might be subjected to more viral contamination, even without direct tissue wetting, compared to fruit sinks where often the xylem functionality is lost towards the end of the season [93].

Furthermore, although the plant mechanisms of viral persistence are poorly understood, DiCaprio et al. [74] observed only few cases of virus detection in strawberry tissues, probably due to antiviral activity by natural metabolites (e.g., polyphenols and phenolic acids) or due to the low pH, while Yang et al. [94] observed, in green onion, a decreasing trend of internalized viruses over time, probably suggesting the presence of inhibitory compounds (e.g., phenolics) inactivating the virus. To date, studies on SARS-CoV-2 and plant internalization/translocation methods are still absent. More research is therefore needed to better understand the possible risks associated with the presence of SARS-CoV-2 in WW or unmonitored water sources when they are used for crop irrigation. However, current researchers' knowledge on enteric viruses, the intrinsic characteristics of SARS-CoV-2 with its limited persistence in the environment (e.g., water and soil), the use of suitable water for irrigation, and the application of appropriate irrigation systems are all elements that could potentially help the community to avoid crop viral contaminations in the future, thus preserving food safety.

## 6. Conclusions and Future Research Directions

The COVID-19 outbreak has spread quickly and unexpectedly worldwide, unfortunately causing the loss of many human lives. The scientific community has found evidence of viable SARS-CoV-2 RNA in WW. Recent attempts to explicate the number of infected persons in the community and assist public health surveillance have relied on detecting SARS-CoV-2 in WW using molecular procedures able to identify genetic material (RNA), though this way does not assess SARS-CoV-2 viability or infectivity. Preliminary research activities have highlighted the absence of a standardized methodological approach for SARS-CoV-2 detection and quantification in WW. However, there is presently no epidemiological proof that WW is a route of transmission, but further investigations are needed, particularly in areas with inadequate sanitation and limited access to drinking water. The standard WW treatment procedures reveal a broad variability in complete removal of viruses from WW, since not all WWTPs have been designed for water reclamation purposes, thereby representing a serious risk to human health and the environment. The fate of SARS-CoV-2 in WWTPs and its removal through the different treatment stages (e.g., primary, secondary, and tertiary treatments) remain unexplored and need urgent consideration, especially where treated WW is utilized as reclaimed water. The few studies reported in the literature show that SARS-CoV-2 RNA is present not only in raw WW, but also in treated WW, mainly when disinfection is inefficient or absent.

However, natural streams and rivers, and in some cases also groundwater, can receive waters other than WWTP effluents (i.e., through combined sewer overflows, run-off from urban and peri-urban catchments, etc.), which can also contain SARS-CoV-2. However, the

current results indicate that the infectivity of virus in river water was null, and therefore the risk is very low, while there is no evidence that SARS-CoV-2 can be present in groundwater.

From an accurate assessment of the literature published to date, it emerged that, despite the fact that SARS-CoV-2 might reach soils intended for agricultural crops, there is a lack of studies demonstrating its survival in soil and eventual internalization or persistence in crops. However, previous studies have proved the presence of other viruses on plant surfaces when direct contact between water droplets (from aspersion systems that use treated or untreated WW) and plant tissues occurred. The application of drip and subsurface irrigation practices corresponds to a useful strategy to limit the viral contamination of comestible products, as in these methods, water typically does not come into direct contact with the plant canopy. Therefore, the selection of the most suitable irrigation method could help in reducing possible risks of food contamination by SARS-CoV-2.

The urgency to address public health-related issues due to the safe and sustainable reuse of water with a certain quality calls for research on some of the following critical points:

- Development of standardized methodological protocols and implementation of cost-effective methods for SARS-CoV-2 identification and estimation in WW.
- Assessment of viral infectivity and survival rate of SARS-CoV-2 in stool, surface water, WW, and other matrices in distinct environmental contexts under different conditions (i.e., temperature, pH, humidity, etc.).
- Study of the role of WWTP units for SARS-CoV-2 deactivation and detection or application of novel remediation technologies.
- Investigation on the effect that disinfected WWTP effluents can have on the ecosystems of streams, rivers, and groundwaters.
- Establishing whether genetic material is present in receiving water bodies, as intact virus particles or as free nucleic acids, as well as infectivity.
- Establishment of appropriate methods for SARS-CoV-2 material sampling (e.g., soil, plant tissue), concentration, quantification, and survival.
- Evaluation of potential transmission of SARS-CoV-2 from water to agricultural crops after their irrigation, and possible risks related to the consumption of contaminated food.
- Implementation of micro-irrigation technologies, which can safely irrigate agricultural crops without bringing fresh produce into direct contact with WW.
- Investigation on the soil–root internalization processes in the case of the presence of SARS-CoV-2.

In conclusion, the current pressure of the pandemic on researchers, who are trying to broaden their knowledge and remedy this critical situation, will allow the re-evaluation of the operation of WW systems and reuse practices, enhancing the role of WWTPs during the COVID-19 pandemic and addressing future health and environmental challenges.

**Author Contributions:** Conceptualization, G.M., G.D.P. and A.T. Funding acquisition, A.T. Methodology, G.M. and G.D.P. Supervision, B.M and A.T. Writing–original draft G.M., G.D.P. and S.L. Writing–review editing G.M., G.D.P., S.L., B.M. and A.T. All authors have read and agreed to the published version of the manuscript.

**Funding:** This study was carried out within the project "Safe and Sustainable Solutions for the Integrated Use of Non-Conventional Water Resources in the Mediterranean Agricultural Sector (FIT4REUSE)" which has received funding from the Partnership on Research and Innovation in the Mediterranean Area (PRIMA) under grant agreement No 1823 (https://fit4reuse.org/). PRIMA is supported by the European Union's Horizon 2020 research and innovation program.

**Institutional Review Board Statement:** Not applicable.

**Informed Consent Statement:** Not applicable.

**Data Availability Statement:** Not applicable.

**Acknowledgments:** Not applicable.

**Conflicts of Interest:** The authors declare no conflict of interest.

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
