# Peer review of "SARS-CoV-2 from Urban to Rural Water Environment: Occurrence, Persistence, Fate, and Influence on Agriculture Irrigation. A Review"

_water, doi:10.3390/w13060764_

Round 1

Reviewer 1 Report

Dear authors,

the paper reviews recent studies and findings about the presence of SARS-CoV-2 gene fragments in wastewater and discusses the relevance in the context of agricultural irrigation. The topic is of relevance. Although not many studies have reported on the presence of SARS-CoV-2 in irrigation water and in irrigated crops. Hence, the basis for the rview is rather thin.

However, the authors appropriately summarise many recent findings and put them into context. The presence of SARS-CoV-2 gene fragments in irrigation water may be of concern, but current findings do not indicate a particular infection risk via this route. Further reserearch needs are pointed out correctly.

The paper needs a proper language polishing and can be published then.

Author Response

Dear authors,

the paper reviews recent studies and findings about the presence of SARS-CoV-2 gene fragments in wastewater and discusses the relevance in the context of agricultural irrigation. The topic is of relevance. Although not many studies have reported on the presence of SARS-CoV-2 in irrigation water and in irrigated crops. Hence, the basis for the rview is rather thin.

However, the authors appropriately summarise many recent findings and put them into context. The presence of SARS-CoV-2 gene fragments in irrigation water may be of concern, but current findings do not indicate a particular infection risk via this route. Further reserearch needs are pointed out correctly.

The paper needs a proper language polishing and can be published then.

Thanks for your comments. From our study it has emerged that in the literature there are currently a few studies on the fate of SARS-CoV-2 from urban to rural water environment, and specifically on its possible presence in crops after they have been irrigated, although some studies report the occurrence of SARS-CoV-2 in soils.

We agree with you that it seems that the risk of detecting SARS-CoV-2 in crops, and thus in food, is low. However, we believe also that more detailed studies are needed to confirm this, and we hope that our study is a starting point for understanding what could be investigated in the future.

Concerning the language, the paper has been revised completely and the English has been edited as requested. Changes have been reported in red color.

Reviewer 2 Report

The manuscript is well organized and sufficiently describes the known knowledge about the risk of possible SARS-CoV-2 transmission from the aquatic environment. It is suitable for publication in Water after revisions as indicated as follows:

Table 1. is confusing and not adequately described in the text.

The paragraph (376-380) is questionable.

Citation of no peer-reviewed studies is not appropriate.

Author Response

The manuscript is well organized and sufficiently describes the known knowledge about the risk of possible SARS-CoV-2 transmission from the aquatic environment. It is suitable for publication in Water after revisions as indicated as follows:

Table 1. is confusing and not adequately described in the text.

Thanks for your note. We agree with you. In the revised version of the manuscript, Table 1 has been edited as suggested. Furthermore, it has been discussed in more detail throughout the text.

The paragraph (376-380) is questionable.

Thanks for your note. It is difficult to respond to this comment without stating a reason. However, in the revised version of the manuscript, the paragraph has been rephrased aiming thus to make clearer what is stated in the text.

Citation of no peer-reviewed studies is not appropriate.

Thanks for your comment. All the no peer-reviewed studies have been removed.